# The synchronicity of COVID-19 disparities: Statewide epidemiologic trends in SARS-CoV-2 morbidity, hospitalization, and mortality among racial minorities and in rural America

**Brian E. Dixon**[1,2]*, **Shaun J. Grannis**[2,3], **Lauren R. Lembcke**[4], **Nimish Valvi**[2], **Anna R. Roberts**[4], **Peter J. Embi**[2,5]

**1** Department of Epidemiology, Fairbanks School of Public Health, Indiana University, Bloomington, Indiana, United States of America, **2** Center for Biomedical Informatics, Regenstrief Institute, Indianapolis, Indiana, United States of America, **3** Department of Family Medicine, School of Medicine, Indiana University, Indianapolis, Indiana, United States of America, **4** Research Data Services, Regenstrief Institute, Indianapolis, Indiana, United States of America, **5** Department of Medicine, School of Medicine, Indiana University, Indianapolis, Indiana, United States of America

* bedixon@regenstrief.org

**Data Availability Statement:** All data used in this study are available from the IUPUI DataWorks repository (https://doi.org/10.7912/D2/26).

## Abstract

### Background

Early studies on COVID-19 identified unequal patterns in hospitalization and mortality in urban environments for racial and ethnic minorities. These studies were primarily single center observational studies conducted within the first few weeks or months of the pandemic. We sought to examine trends in COVID-19 morbidity, hospitalization, and mortality over time for minority and rural populations, especially during the U.S. fall surge.

### Methods

Data were extracted from a statewide cohort of all adult residents in Indiana tested for SARS-CoV-2 infection between March 1 and December 31, 2020, linked to electronic health records. Primary measures were per capita rates of infection, hospitalization, and death. Age adjusted rates were calculated for multiple time periods corresponding to public health mitigation efforts. Comparisons across time within groups were compared using ANOVA.

### Results

Morbidity and mortality increased over time with notable differences among sub-populations. Initially, hospitalization rates among racial minorities were 3–4 times higher than whites, and mortality rates among urban residents were twice those of rural residents. By fall 2020, hospitalization and mortality rates in rural areas surpassed those of urban areas, and gaps between black/brown and white populations narrowed. Changes across time among demographic groups was significant for morbidity and hospitalization. Cumulative morbidity and mortality were highest among minority groups and in rural communities.

**Funding:** BED receives funding from the U.S. Agency for Healthcare Research and Quality (R21HS025502) to study health information exchange. The Regenstrief Institute as well as the IU Fairbanks School of Public Health received funds from the State of Indiana and Marion County Public Health Department to support COVID-19 response and mitigation, including disease surveillance and outcomes measurement. The funders did not have influence over the findings or this publication.

**Competing interests:** The authors have declared that no competing interests exist.

## Conclusions

The synchronicity of disparities in COVID-19 by race and geography suggests that health officials should explicitly measure disparities and adjust mitigation as well as vaccination strategies to protect those sub-populations with greater disease burden.

## Introduction

The rapid spread of coronavirus disease 2019 (COVID-19), caused by the severe acute respiratory syndrome coronavirus 2 (SARS-CoV-2) virus, has challenged health systems. As of May 5, 2021, more than 155 million individuals were infected, and over 3.2 million individuals have died from COVID-19 globally [1]. In the United States, there are over 32 million cases, 2.1 million hospitalizations, and 580,000 deaths associated with COVID-19. At the end of 2020, the US Centers for Disease Control and Prevention (CDC) reported a COVID-19-associated hospitalization rate of 326.7 per 100,000 population [2].

Data from China, Italy, the US, and other nations suggest that hospitalization and mortality are associated with age as well as gender, in which older and male populations are at higher risk of severe outcomes including death [3–5]. Moreover, early evidence in the US identified unequal patterns in hospitalization and mortality from COVID-19 in dense urban environments with respect to race and ethnicity [6, 7]. Existing studies, however, are limited with respect to scope, data, and temporality. Early evidence largely comes from single, urban center studies or regional data during the first wave [8, 9] of the COVID-19 pandemic. Subsequent waves, or surges, have not been examined. For example, it is unclear whether the same populations were impacted in fall versus the spring 2020. Furthermore, most studies examine patients after inpatient admission. There are limited studies on individuals with COVID-19 in community settings. For example, early data suggested that testing rates per capita were unequal in the US with respect to race and ethnicity [10], yet it is unclear how these rates have changed over time. Early studies from April 2020 also identified unequal infections among racial groups [11, 12]. However, these studies used pre-pandemic, area-level indicators to associate the proportion of Black residents with overall cases reported for the same area. These studies did not use data at the individual level to examine infections, hospitalizations, or deaths.

There further exists limited evidence on individuals in rural communities, with few studies that compare rural patients to those in urban settings. Nearly 1-in-5 Americans live in a rural county [13], which are often labeled as medically underserved areas. Using data from CDC gathered two years before the pandemic, Kaufman et al. [14] estimate rural residents to be at increased risk of hospitalization and death from COVID-19. A similar analysis using data from 2018 by Peters [15] concluded many rural areas are vulnerable to COVID-19 given populations with advanced age and high rates of death due to conditions such as cardiovascular disease. In a single site study in rural Georgia, individuals hospitalized with COVID-19 from March to May 2020 found that, despite a higher burden of comorbid conditions, both critical care and in-hospital mortality was lower than in New York City, as well as China and Italy [16]. A national, population-based study similarly found lower rates of COVID-19 mortality in rural areas compared to urban areas using county-level data through June 12, 2020 [17]. Given limited data from rural communities through Spring 2020, additional analyses are warranted.

No studies to date examine the resurgence of COVID-19 infections within the US during the fall, and little is known about shifts in morbidity and mortality among population groups

over time. This study examines the epidemiologic trends in COVID-19 infection, hospitalization, and death rates in Indiana with a focus on health equity. The study examines a large, statewide cohort, including individuals tested by local and state health departments. It further examines care delivered at a wide array of settings, including critical access hospitals and rural county hospitals. Per capita rates, when stratified by age, race, sex, and geography, can shed light on changes in morbidity and mortality over time, as well as within and among sub-populations. Understanding patterns of COVID-19 morbidity and mortality beyond individual health systems and major metropolitan areas can inform national strategies to mitigate the ongoing spread of COVID-19, including vaccination strategies that seek to immunize based primarily on age.

## Methods

### Setting

The setting for this study is the State of Indiana, which is the 16[th] largest state in the US with respect to population density and 38[th] by area. The state has a growing population of 6,732,219 individuals, of which 5,063,133 (75.2%) are adults. Most residents identify as non-Hispanic White (78.4%), followed by Black or African American (9.9%), Hispanic (7.8%), and Asian (2.6%). Approximately 21.7% of residents reside in a rural county.

The State of Indiana reported its first case of COVID-19 on March 6, 2020. Similar to many locations, Indiana implemented public health interventions, including a stay-at-home order, to mitigate spread of COVID-19. On May 1, 2020, following declining rates of hospitalization for COVID-19, the Governor ended the stay-at-home order and initiated a phased re-opening plan [18]. New cases increased while hospitalizations declined into the summer, flattening until a second wave began following Labor Day. The second wave consisted of steady increases in new cases as well as hospitalizations and deaths, all of which climbed through the holidays before leveling off towards the end of the year.

### Data and sources

We use data from multiple sources integrated into the Regenstrief Institute COVID-19 Dashboard [19], a data visualization tool developed in response to the pandemic that leverages clinical and administrative health data from the Indiana Network for Patient Care (INPC) [20]. The INPC is one of the nation's largest health information networks, which includes 38 distinct health systems representing more than 100 hospitals, commercial laboratories, and physician practices across Indiana [21]. The INPC further includes COVID-19 test results from the Indiana Department of Health (IDOH), which receives test results from large commercial labs contracted for pandemic response as well as local health departments which perform strategic testing in communities identified as high risk, such as nursing homes, prisons, and homeless shelters. All testing data, regardless of source, are linked to hospitalization data as well as death records from IDOH. The combined data represent >95% of the 5 million adult residents who interact with the state's health system.

We extracted data on all adults (age ≥18 years) tested for COVID-19 in the health system or community, as well as those diagnosed with COVID-19 during a clinical encounter. For each individual, we queried the following information from the INPC: COVID-19 test results, age, sex, race, hospitalizations up to 21 days before or after a positive COVID-19 test, and geography associated with home address. Hospitalizations before positive diagnosis were included due to delays in testing, especially at the start of the epidemic. During March and April 2020, most patients infected with the SARS-CoV-2 virus were admitted with COVID-like symptoms before testing positive. All positive cases were identified using RT-PCR tests

recorded in medical records or reported to the public health department, including community-based testing efforts statewide by public health authorities. Individuals testing positive were only counted once, during the period of their first positive result. All patient addresses were geocoded using an established method [22] with rurality determined by a classification system developed by Purdue University for Indiana's geography based on ZIP Code [23].

The fully identified dataset involving linked medical records, death records, and COVID-19 testing results was extracted and managed by the trained data analysts (LRR and ARR) as the Regenstrief Data Services team functions as the honest data broker for the INPC. These data were fully de-identified prior to release to the study team for analysis. The de-identified data used for analysis are available for reproduction and secondary use through IUPUI Data-Works, a repository for research data for faculty at Indiana University-Purdue University Indianapolis [24]. The dataset DOI is https://doi.org/10.7912/D2/26.

### Data analysis

We used epidemiological methods to calculate descriptive statistics, including rates per 100,000 population, also known as per capita rates. These rates provide an objective method for comparing population characteristics when communities or groups vary in size. Denominator data for calculating per capita rates came from 2018 U.S. Census estimates. All rates were age-adjusted using American Community Survey estimates.

Statistics were calculated overall and for multiple time periods corresponding to the state's initial lockdown and subsequent re-opening plan. We examined data from the start of the Indiana epidemic (March 6, 2020) thru the end of the stay-at-home order (April 30, 2020), referred to as Phase 1. Following the stay-at-home order, Indiana initiated a staged reopening. Each subsequent stage reopened additional sectors of the economy or expanded capacity in a given sector. Full details of each stage can be found on the Governor's Back on Track website [18]. The reopening occurred from May 1, 2020 through September 7, 2020 (Labor Day), when the only remaining restrictions included a statewide mask ordinance and a restriction on gatherings larger than 250 people. This is referred to as Phase 2. Finally, we examined data from September 8, 2020 through December 31, 2020, referred to as Phase 3.

We further investigated differences across study Phases and per-capita rates by each of the demographic variables using analysis of variance (ANOVA). All statistical comparisons were two-sided and a p-value <0.05 was considered statistically significant. Analyses were performed using SAS version 9.4 (SAS Institute Inc., Cary, NC, USA). Institutional review board approval for the study was obtained from Indiana University. The IRB further waived the requirement for informed consent as the research involved previously collected medical record data from individuals tested for COVID-19 by public health or health care providers.

## Results

Through December 31, 2020, a total of 1,833,218 unique, adult Indiana residents were tested for COVID-19, which accounts for 36.2% of the statewide adult population. Of those tested, 354,539 (19.3%) unique individuals tested positive for COVID-19 infection. Among those infected, 31,352 (8.8%) were hospitalized. A total of 8,104 (0.2% of infected individuals) died either during their hospital course or at home following COVID-19 infection.

### Overall COVID-19 infection and burden

Characteristics, as well as morbidity and mortality, of individuals tested for COVID-19 in Indiana are summarized in **Table 1**. More women than men per capita were tested and positive for COVID-19, yet men had higher hospitalizations and mortality per capita compared to women.

**Table 1. Characteristics and rates for Indiana residents tested for, infected with, hospitalized with, and death following infection from COVID-19 through December 31, 2020; State of Indiana.**

| Characteristics | Individuals Tested for COVID-19 | | | COVID-19 Cases | | | COVID-19 Hospitalizations | | | COVID-19 Deaths | | |
|---|---|---|---|---|---|---|---|---|---|---|---|---|
| | N | % | rate per capita* | N | % | rate per capita* | N | % | rate per capita* | N | % | rate per capita* |
| Total | 1833218 | | 36207.2 | 354539 | | 7002.4 | 31352 | | 619.2 | 8104 | | 160.1 |
| **Gender** | | | | | | | | | | | | |
| Female | 1031547 | 56.3% | 39709.7 | 190632 | 53.8% | 7338.4 | 15890 | 50.7% | 611.7 | 4008 | 49.5% | 154.3 |
| Male | 790687 | 43.1% | 32071.2 | 162363 | 45.8% | 6585.6 | 15459 | 49.3% | 627.0 | 4073 | 50.3% | 165.2 |
| **Age Category** | | | | | | | | | | | | |
| 18–19 | 73702 | 4.0% | 39707.1 | 14169 | 4.0% | 7633.6 | 130 | 0.4% | 70.0 | 2 | 0.0% | 1.1 |
| 20–29 | 357306 | 19.5% | 39038.2 | 70064 | 19.8% | 7655.0 | 1472 | 4.7% | 160.8 | 38 | 0.5% | 4.2 |
| 30–39 | 312253 | 17.0% | 37227.7 | 58867 | 16.6% | 7018.3 | 1950 | 6.2% | 232.5 | 75 | 0.9% | 8.9 |
| 40–49 | 284677 | 15.5% | 34644.3 | 59479 | 16.8% | 7238.4 | 2771 | 8.8% | 337.2 | 154 | 1.9% | 18.7 |
| 50–59 | 294838 | 16.1% | 32976.5 | 58540 | 16.5% | 6547.5 | 4731 | 15.1% | 529.1 | 408 | 5.0% | 45.6 |
| 60–69 | 258639 | 14.1% | 34580.4 | 46261 | 13.0% | 6185.2 | 6756 | 21.5% | 903.3 | 1244 | 15.4% | 166.3 |
| 70–79 | 157831 | 8.6% | 38123.8 | 27605 | 7.8% | 6667.9 | 7241 | 23.1% | 1749.1 | 2038 | 25.1% | 492.3 |
| 80+ | 93972 | 5.1% | 38238.7 | 19554 | 5.5% | 7956.8 | 6301 | 20.1% | 2564.0 | 4145 | 51.1% | 1686.7 |
| **Race** | | | | | | | | | | | | |
| White | 1420211 | 77.5% | 32772.9 | 272622 | 76.9% | 6291.1 | 24140 | 77.0% | 557.1 | 6538 | 80.7% | 150.9 |
| African American | 164730 | 9.0% | 37075.1 | 31596 | 8.9% | 7111.2 | 4922 | 15.7% | 1107.8 | 952 | 11.7% | 214.3 |
| Asian | 29043 | 1.6% | 23872.1 | 5638 | 1.6% | 4634.2 | 353 | 1.1% | 290.2 | 54 | 0.7% | 44.4 |
| American Indian / Native Alaskan | 3961 | 0.2% | 35423.0 | 1193 | 0.3% | 10668.9 | 69 | 0.2% | 617.1 | 4 | 0.0% | 35.8 |
| Native Hawaiian / Pacific Islander | 3057 | 0.2% | 142783.7 | 783 | 0.2% | 36571.7 | 79 | 0.3% | 3689.9 | 19 | 0.2% | 887.4 |
| Other or Unknown | 212216 | 11.6% | 194470.6 | 42707 | 12.0% | 39135.9 | 1789 | 5.7% | 1639.4 | 537 | 6.6% | 492.1 |
| Geography | | | | | | | | | | | | |
| Rural | 379567 | 20.7% | 34585.2 | 82276 | 23.2% | 7496.8 | 6962 | 22.2% | 634.4 | 1890 | 23.3% | 172.2 |
| Urban | 1453651 | 79.3% | 36656.1 | 272263 | 76.8% | 6865.5 | 24390 | 77.8% | 615.0 | 6214 | 76.7% | 156.7 |

*Rate per capita adjusted for age

With respect to age, testing and morbidity were highest in the younger (0–29) and older (80+) groups. Hospitalization and mortality per capita increased with age, with older (70+) populations possessing significantly higher hospitalizations and deaths per capita. With respect to race, morbidity, hospitalization, and mortality rates per capita were higher among racial minority groups, especially Native Hawaiian/Pacific Islanders, American Indian/Native Alaskans, and African Americans respectively. Individuals who did not report their race during testing or hospitalization also possessed high rates of morbidity and mortality. With respect to geography, urban residents were tested more frequently. However, per capita morbidity, hospitalization, and mortality were highest among rural populations.

## Comparison of infection, hospitalization, and death over time

Table 2 summarizes per capita infections, hospitalizations, and deaths during the three phases of the COVID-19 epidemic in Indiana through the end of 2020. Infections, hospitalizations, and deaths per capita all increased over time. Hospitalization and death increased with age and increased across phases for all groups. There are notable differences among sub-populations.

**Table 2. Comparison of per capita rates for COVID-19 infection, hospitalization and death for residents across three phases of the epidemic; State of Indiana.**

| Characteristics | COVID-19 Cases | | | COVID-19 Related Hospitalization | | | COVID-19 Related Deaths | | |
|---|---|---|---|---|---|---|---|---|---|
| | Phase 1 (rate per capita) | Phase 2 (rate per capita) | Phase 3 (rate per capita) | Phase 1 (rate per capita) | Phase 2 (rate per capita) | Phase 3 (rate per capita) | Phase 1 (rate per capita) | Phase 2 (rate per capita) | Phase 3 (rate per capita) |
| Total | 402.8 | 1375.2 | 5224.4 | 84.6 | 133.3 | 402.5 | 41.5 | 41.1 | 77.5 |
| **Gender** | | | | | | | | | |
| Female | 419.4 | 1421.9 | 5497.2 | 80.3 | 134.5 | 398.2 | 39.8 | 41.9 | 72.6 |
| Male | 377.3 | 1308.1 | 4900.3 | 89.1 | 132.1 | 406.8 | 42.5 | 40.1 | 82.6 |
| | | | p = 0.0035[*] | | | p = 0.0004[*] | | | p = 0.02[*] |
| **Age Category** | | | | | | | | | |
| 18–19 | 83.5 | 2003.1 | 5547.0 | 5.4 | 23.2 | 42.0 | 0.0 | 0.0 | 0.2 |
| 20–29 | 265.2 | 1858.7 | 5531.4 | 15.6 | 46.5 | 98.9 | 1.1 | 1.5 | 1.5 |
| 30–39 | 370.3 | 1377.3 | 5270.7 | 35.2 | 62.1 | 135.9 | 2.3 | 2.7 | 3.9 |
| 40–49 | 426.8 | 1379.4 | 5432.3 | 58.2 | 89.9 | 190.1 | 5.1 | 5.7 | 7.9 |
| 50–59 | 416.2 | 1144.5 | 4986.8 | 88.7 | 119.0 | 322.0 | 13.3 | 12.0 | 20.4 |
| 60–69 | 396.7 | 1016.1 | 4772.5 | 126.1 | 181.2 | 598.2 | 43.7 | 39.8 | 82.8 |
| 70–79 | 480.7 | 1084.1 | 5103.4 | 199.5 | 330.7 | 1221.0 | 116.9 | 125.4 | 250.0 |
| 80+ | 1026.6 | 1501.5 | 5428.7 | 323.5 | 502.1 | 1743.2 | 446.8 | 436.6 | 803.3 |
| | | | p < .0001[*] | | | p = 0.016[*] | | | p = 0.14[*] |
| **Race** | | | | | | | | | |
| White | 287.2 | 1068.1 | 4935.9 | 62.4 | 109.2 | 386.5 | 35.3 | 36.5 | 79.1 |
| African American | 926.4 | 1816.3 | 4368.5 | 285.8 | 287.6 | 536.1 | 89.1 | 63.0 | 62.1 |
| Asian | 433.2 | 922.2 | 3278.8 | 51.0 | 87.1 | 152.1 | 14.8 | 10.7 | 18.9 |
| American Indian / Native Alaskan | 3559.3 | 2012.2 | 5097.5 | 232.5 | 152.0 | 241.5 | 8.9 | 0.0 | 26.8 |
| Native Hawaiian / Pacific Islander | 2942.6 | 10462.4 | 23166.7 | 747.3 | 747.3 | 2195.2 | 186.8 | 280.2 | 420.4 |
| Other or Unknown | 2607.1 | 12557.2 | 23971.6 | 188.8 | 549.8 | 904.5 | 137.5 | 184.2 | 170.4 |
| | | | p = 0.02[*] | | | p = 0.04[*] | | | p = 0.22[*] |
| **Geography** | | | | | | | | | |
| Rural | 355.9 | 1232.6 | 5908.2 | 51.6 | 124.5 | 459.7 | 24.7 | 43.7 | 103.8 |
| Urban | 415.8 | 1414.6 | 5035.2 | 93.7 | 135.8 | 386.6 | 46.1 | 40.4 | 70.2 |
| | | | p = 0.01[*] | | | p = 0.02[*] | | | p = 0.19[*] |

**Note:** *p* values for demographic groups and for the three phases were calculated using ANOVA.

[*]*p* values indicate comparisons over the three phases.

[a] Phase 1 (March–April, 2020)

[b] Phase 2 (May 1, 2020 –September 7, 2020)

[c] Phase 3 (September 8, 2020 –December 31, 2020)

**COVID-19 infection rates (morbidity).** Infection rates increased for both men and women over the study period. The rates were higher among females, which increased from 419.4 per 100,00 (phase 1) to 5497.2 per 100,000 (phase 3). At the start of the pandemic, young adults (18 to 19 years) had the lowest infection rates at 83.5 per 100,000 (phase 1), which increased later to 5547.0 per 100,000 (phase 3). The oldest adults (≥80 years) possessed the highest infection rates across the three phases, increasing from 1026.6 per 100,000 (phase 1) to 5428.7 per 100,000 (phase 3). With respect to race, rates during Phase 1 were lowest among Whites (287.2 per 100,000), while African American (926.4 per 100,000) and American Indian/Native Alaskans (3559.3 per 100,000) had higher rates. Infection rates significantly

increased by phase for all demographic characteristics like sex (p = 0.0035), age-group (p <
.0001), race (p = 0.02), and geography (p = 0.01) during the study period.

**COVID-19 hospitalization rates.** Hospitalization rates for men increased from 89.1 per
100,000 (phase 1) to 406.8 per 100,000 (phase 3), and they were higher among men in phase 1
and phase 3. Hospitalization rates were highest among the oldest adults (≥80 years) across the
three phases with a low of 323.5 per 100,000 (phase 1) to 1743.2 per 100,000 (phase 3). Rates
were lowest among young adults (18 to 19 years) 5.4 per 100,00 (phase 1), which increased to
42.0 per 100,000 (phase 3). Rates were highest for Native Hawaiian/Pacific Islanders (747.3 per
100,000; phase 1 and 2) and among African-Americans (285.8 per 100,000; phase 1), which
increased to 2195.2 per 100,000 (phase 3) and 536.1 per 100,000 (phase 3), respectively. For
rural and urban populations, hospitalization rates were almost 2 times higher in the urban
population 93.7 per 100,000 during Phase 1. However, hospitalization rates increased in the
rural population from 51.6 per 100,000 (phase 1) to 459.7 per 100,000 population (phase 3).
Hospitalization rates were significantly different across the phases by sex (p = 0.0004), age-
group (p = 0.016), race (p = 0.04), and geography (p = 0.02).

**COVID-19 mortality rates.** Mortality rates increased for both men and women during
the study period. For men, mortality increased from 42.5 per 100,000 (phase 1) to 82.6 per
100,000 (phase 3), while for women rates increased from 39.8 per 100,000 (phase 1) to 72.6 per
100,000 (phase 3). Mortality progressively increased among adults 60 years and older, with the
highest rates observed among the oldest adults (≥ 80 years), increasing from 446.8 per 100,000
(phase 1) to 803.3 per 100,000 during Phase 3. Mortality was highest among Native Hawaiian/
Pacific Islanders, which increased from 186.8 per 100,000 (phase 1) to 420.4 per 100,000
(phase 3). Mortality among African-Americans progressively reduced from 89.1 per 100,000
(phase 1) to 62.1 per 100,000 (phase 3), while among Whites it increased from 35.3 per 100,000
(phase 1) to 79.1 per 100,000 (phase 3). Mortality remained low in the rural population in the
early phase, 24.7 per 100,000 (phase 1), but increased to 103.8 per 100,000 in Phase 3. In urban
areas, mortality increased from 46.1 per 100,000 population (phase 1) to 70.2 per 100,000
(phase 3). Mortality rates across the demographics by phase differed only for sex (p = 0.02),
and rates were not statistically significant for other demographic characteristics.

**Fig 1** summarizes per capita rates for hospitalizations and mortality, stratified by race, in
each phase. Hospitalizations were higher for African Americans in all phases. Mortality was
higher for African Americans in the first two phases. Morbidity and mortality were highest in
every phase among those who either reported their race as Other or did not disclose their race
at the time of testing or treatment.

Morbidity and hospitalizations were higher for urban populations during the first two
phases. Deaths were higher among urban populations only in Phase 1. Over time, per capita
morbidity (**Fig 2**) and negative outcomes, hospitalization and mortality (**Fig 3**), shifted to rural
populations. Morbidity shifted in Phase 3 following the conclusion of the state's re-opening
plan. Higher per capita hospitalization and mortality among rural populations began towards
the end of Phase 2 then accelerated in Phase 3.

## Discussion

Among a statewide cohort of individuals tested for COVID-19, we examined epidemiological
trends in testing, infection, hospitalization, and death rates across three time periods corre-
sponding to mitigation efforts by public health authorities. Infections due to the SARS-CoV-2
virus increased over time, yet the impact of COVID-19 was not even across sub-populations.
Following the initial lockdown in the spring, the gap between White and African American
morbidity and mortality narrowed, although burden remained significantly higher among

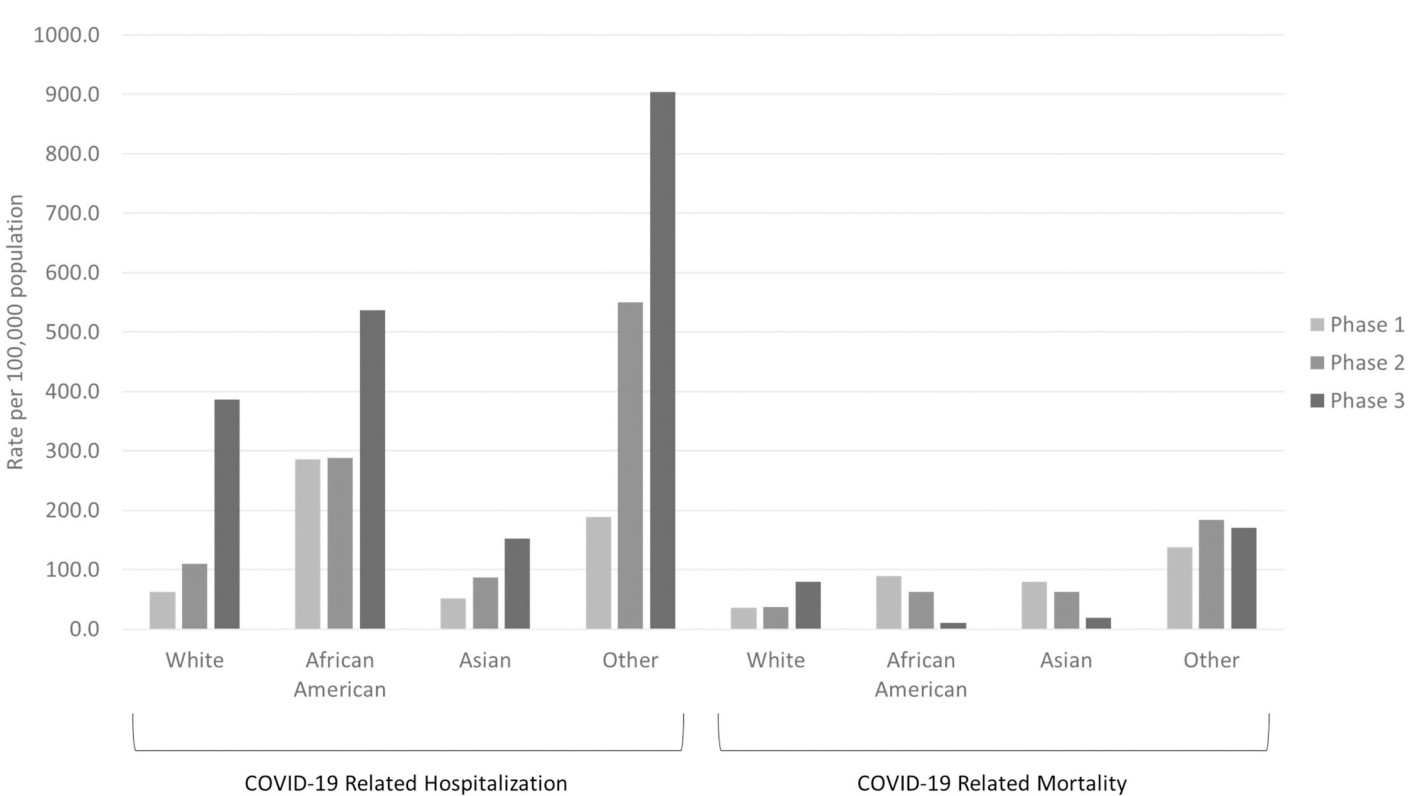

**Fig 1. Per capita rates for hospitalizations and deaths due to COVID-19 among adults in Indiana, stratified by race, during three distinct time periods between March 1 and December 31, 2020.** State of Indiana.

African American populations. As summer turned into fall, burden among rural communities increased and surpassed urban communities through the end of the year. These trends reveal a synchronicity of disparities, or simultaneous set of inequities in outcomes among distinct populations impacted by COVID-19. In other words, Black and Brown populations as well as Rural populations experienced unequal burden and hospitalization due to COVID-19 at the same time. Synchronicities in disparities have implications for continued mitigation of disease spread as well as vaccination strategies.

There are many similarities in the Indiana trends with prior studies as well as national trends. Rates per capita for hospitalization and death increase with age [4, 6, 25–27]. Furthermore, hospitalization and death per capita was greater among men versus women [26, 27], even though women experienced greater morbidity. Moreover, burden of COVID-19 within the African American community overall was much greater than its proportional composition of the state's population [7, 12]. Per capita hospitalization among African Americans grew and remained highest among all sub-populations throughout the pandemic. Burden among American Indian/Native Alaskan and Native Hawaiian/Pacific Islander populations were also among the highest overall and during most time periods, a trend observed nationally [28].

While data in this study share much in common with prior studies, there are several unique characteristics that distinguish our work. First, the study uses a statewide, community dataset of individuals tested for COVID-19 linked to electronic medical records. Testing data includes results from hospital-based, commercial, and public health laboratories creating a comprehensive source measuring testing per capita. Furthermore, our data source contains statewide hospitalization and death data from 38 distinct health systems. Second, the study measures

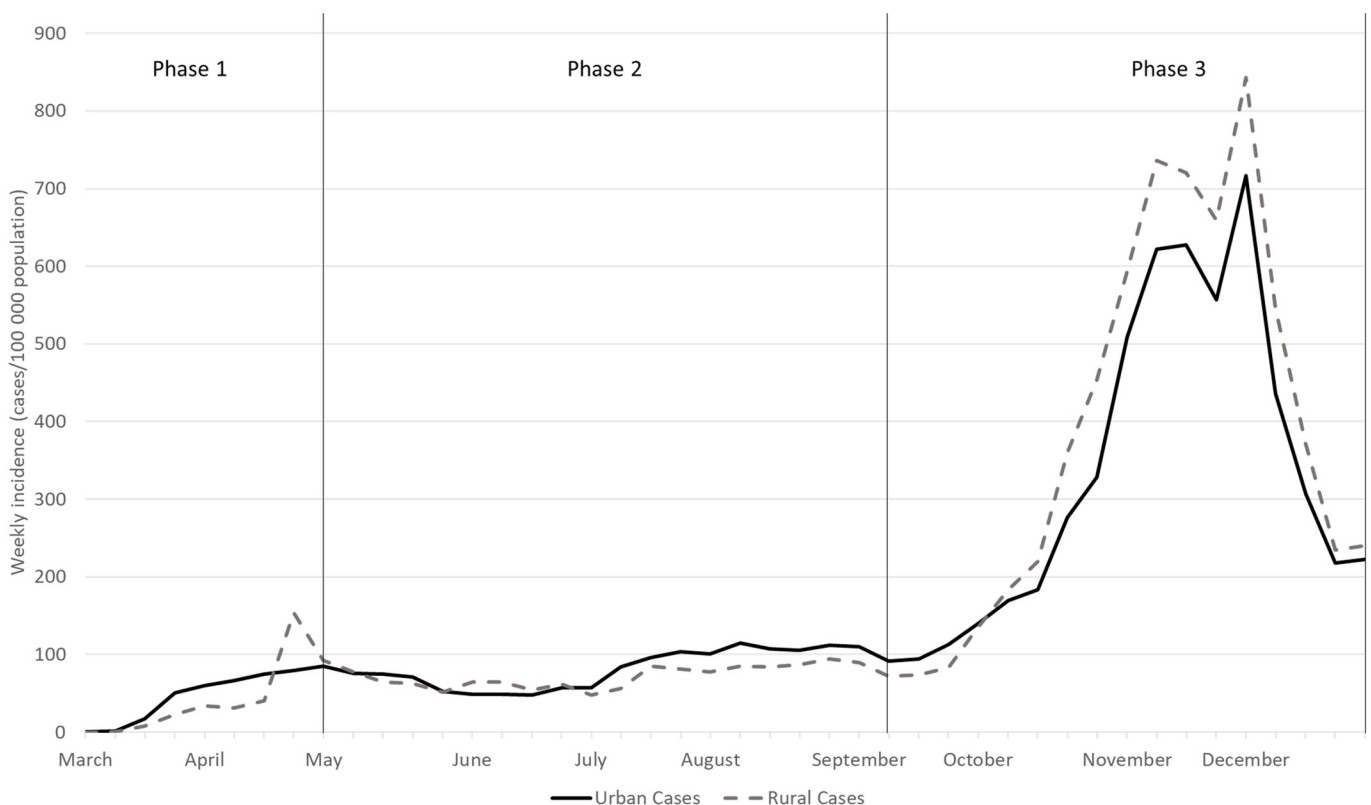

**Fig 2. Weekly COVID-19 incidence, defined as cases per 100,000 population, among adults in Indiana, stratified by urban versus rural county of residence between March 1 and December 31, 2020.** State of Indiana.

burden of disease and outcomes during the fall surge, something few studies to date have reported. Stratification by phase is also unique, allowing comparison of burden and outcomes over time. These methods would not be possible without a robust electronic data infrastructure in Indiana aided by a 16+ year health information exchange [29] that partnered with the state health department, county health departments, and health care systems in response to the COVID-19 pandemic [20]. This multi-sector approach aligns with the vision set forth by the Public Health 3.0 framework described by DeSalvo et al. [30, 31].

Although prior studies document racial disparities, especially during the initial phase of the pandemic, this study presents data on racial disparities in COVID-19 morbidity, hospitalization, and mortality over time. Rates, adjusted for population size, clearly show significant burden on African American populations during each phase of the pandemic. Although deaths per capita were lower than other racial groups in the fall surge, the cumulative mortality is 50% higher than mortality among White populations. Moreover, burden among Native Hawaiian/ Pacific Islander populations, despite accounting for a small percentage of the population, is nearly 10 times that of Whites. Among those who did not disclose or reported their race as 'Other,' hospitalization and death rates are 2–4 times higher than Whites. It is not unreasonable to assume that many Hispanics may be in that group as they may not wish to disclose demographic details and this minority group has been shown to have higher rates of hospitalizations and mortality in national studies [32]. Therefore, we conclude that while racial disparities narrowed later in the pandemic, especially as burden shifted from urban to rural communities, cumulative burden on racial minorities from COVID-19 are severe. This burden

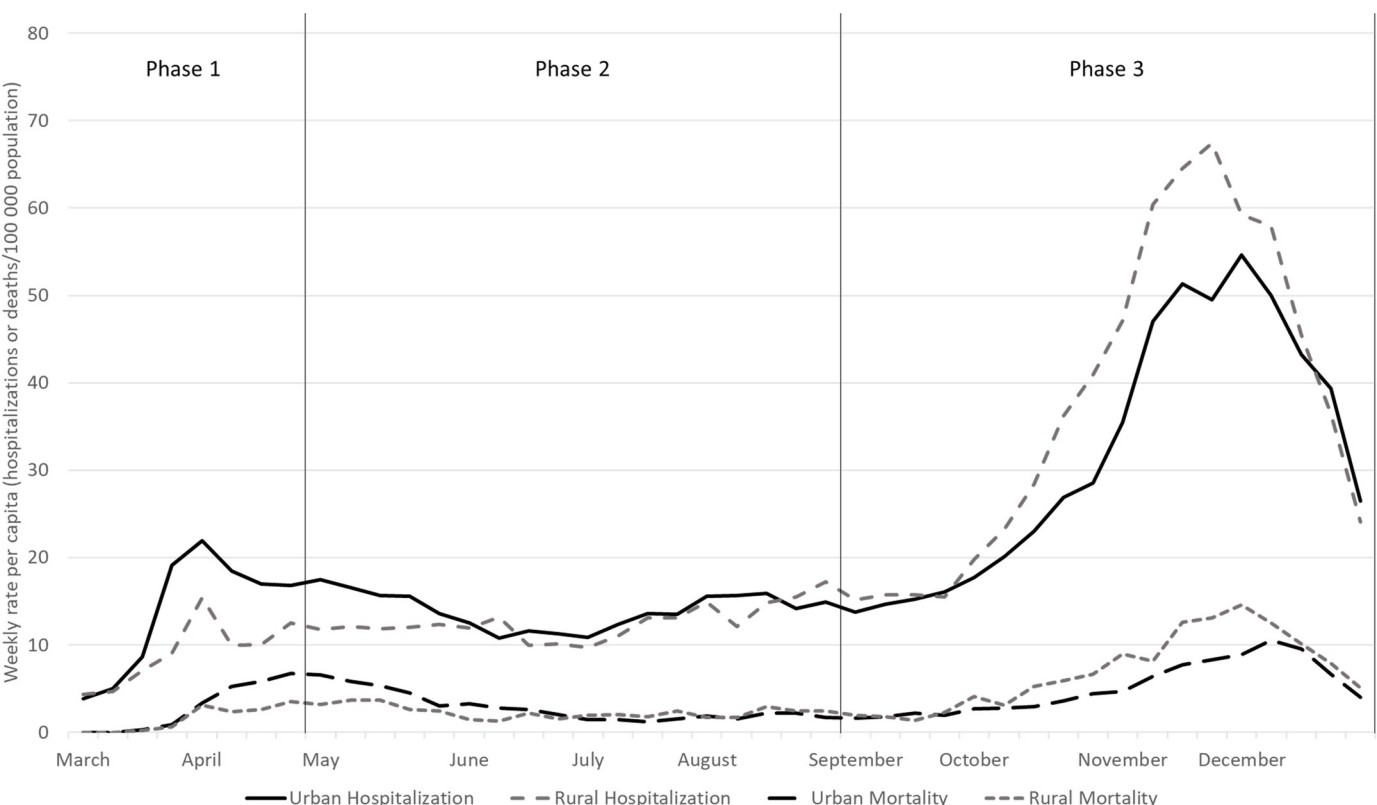

**Fig 3. Weekly rates per capita for hospitalizations and deaths due to COVID-19 among adults in Indiana, stratified by urban versus rural county of residence between March 1 and December 31, 2020.**

exacerbates existing health disparities, necessitating action as the nation attempts to both mitigate further disease spread and protect vulnerable populations through vaccination.

Another distinguishing feature of this analysis is a focus on rural communities. Earlier studies [14, 15, 17] only provided information on risk or observed an initial low impact in rural communities. A brief report on COVID-19 incidence thru October 2020 from the U.S. Centers for Disease Control and Prevention [33] revealed that morbidity shifted nationally from urban to rural areas beginning in late summer. These trends are mirrored in this study. Yet this study further provides evidence on shifting hospitalizations and mortality, which both surpassed urban rates per capita. However, these differences were not statistically significant.

Rural morbidity is of great concern, however, as many rural areas are medically underserved. Hospitals in rural areas may possess few ICU beds, and they may lack the staff necessary to handle an influx of COVID-19 cases [34]. During the fall surge, several rural hospitals in Indiana reached capacity, necessitating transfers to urban centers, which can be more than 2 hours away from the resident's home. This placed additional burden on urban hospitals already managing increased workload and burden from local residents infected with COVID-19. The situation further caused a response from public health in which elective procedures were reduced by order of the Governor, placing financial strain on both rural and urban facilities. More attention is needed on the impact of COVID-19 in rural areas, combined with reasonable policies to support rural mitigation strategies and equitable distribution of vaccines to rural populations.

## Limitations

This observational study has several limitations worth noting. Observational clinical data (e.g., "real-world evidence"), from which much of our findings are derived, is known to have potential biases [35]. First, a significant number of race classifications were reported as Other or Unknown. Similarly, the dataset could not identify ethnicity, as these data are also missing for many patients. Medical records as well as other health information systems, must improve the capture rates for race and ethnicity to enable large scale measurement of health disparities so public health can work with health systems to ensure health for all persons [36, 37]. Second, these data represent hospitalization and mortality among individuals from one state. The patterns observed in Indiana may not generalize to all geographic regions of the U.S. or other countries. These patterns are more applicable to states or regions where the population has similar demographic or rural characteristics.

## Public health implications

This study offers several implications for public health in the wake of the COVID-19 pandemic. First, trends demonstrate a flattening of the curve following the initial stay-at-home order from public health authorities. As the state re-opened, morbidity and mortality increased during subsequent phases. This suggests aggressive mitigation for a longer period of time may be necessary for stronger disease control and prevention. Moreover, sub-population differences highlight the need for more nuanced mitigation policies, such as data-driven approaches that target high-risk groups, that can evolve over time.

As public health attempts to mitigate disease spread going forward, additional attention should be paid to racial minority and rural populations. Testing increased per capita among racial minority groups in Indiana, enabling better detection of morbidity. Equitable testing was not sufficient for stemming hospitalization due to COVID-19. Mortality decreased among minority groups in the latter phases, yet this might be attributable to improved clinical management rather than contact tracing and isolation which our data did not measure. With respect to rural populations, morbidity, hospitalization, and mortality steadily increased over time, suggesting perhaps rural health departments struggled with mitigation strategies or rural populations ignored mask ordinances, restrictions on social gatherings, and/or other public health interventions. Anecdotally, we observed complaints from several rural county health officers that local authorities would not enforce ordinances and that residents overtly refused to comply with many policies. More research is necessary to confirm these observations and support the development of more robust mitigation policies.

Strategies to vaccinate against COVID-19 need to explicitly address racial disparities. Poorer health outcomes among racial minorities is often attributable to lack of health care access [38], including preventative medicine and vaccination. In a majority White state, we achieved equity in testing. This means we can achieve equity in vaccination. However, current policies focus on age and comorbid conditions to drive decisions about which populations should receive vaccines first. While age places individuals at higher risk of hospitalization and death, this study demonstrates that racial minorities and rural populations also should be prioritized given their higher morbidity and mortality. If health departments are serious about addressing social determinants and racial disparities, they must factor these phenomena into vaccination plans.

## Acknowledgments

The authors thank Jack VanSchaik and Connor McAndrews, MS, of the Regenstrief Institute for their assistance with the data management for this analysis. We further acknowledge the

broader COVID-19 Dashboard Team, part of the Regenstrief Data Services group, for their daily efforts to gather, link, and publish data on COVID-related infections, hospitalizations, and deaths for communities across Indiana. Data management during the pandemic has been challenging, and they have risen to every challenge. The authors further thank Nir Menachemi, PhD, of the IU Fairbanks School of Public Health for his feedback on an early draft of this manuscript.

## Author Contributions

**Conceptualization:** Brian E. Dixon, Peter J. Embi.

**Data curation:** Lauren R. Lembcke, Anna R. Roberts.

**Formal analysis:** Brian E. Dixon, Nimish Valvi.

**Funding acquisition:** Peter J. Embi.

**Investigation:** Brian E. Dixon, Shaun J. Grannis.

**Methodology:** Brian E. Dixon, Peter J. Embi.

**Project administration:** Brian E. Dixon, Anna R. Roberts.

**Supervision:** Anna R. Roberts, Peter J. Embi.

**Validation:** Lauren R. Lembcke.

**Visualization:** Shaun J. Grannis, Peter J. Embi.

**Writing – original draft:** Brian E. Dixon.

**Writing – review & editing:** Brian E. Dixon, Shaun J. Grannis, Lauren R. Lembcke, Nimish Valvi, Anna R. Roberts, Peter J. Embi.

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
