## [Decision Letter · Decision Letter 0]

23 Mar 2021

PONE-D-21-05556

The COVID-19 health equity twindemic: Statewide epidemiologic trends of SARS-CoV-2 outcomes among racial minorities and in rural America

PLOS ONE

Dear Dr. Dixon,

Thank you for submitting your manuscript to PLOS ONE. After careful consideration, we feel that it has merit but does not fully meet PLOS ONE’s publication criteria as it currently stands. Therefore, we invite you to submit a revised version of the manuscript that addresses the points raised during the review process.

Please respond to the reviewer comments on a point-by-point basis and revise the manuscript accordingly. 

We look forward to receiving your revised manuscript.

Kind regards,

Jeffrey Shaman

Academic Editor

PLOS ONE

Journal Requirements:

Reviewers' comments:

Reviewer's Responses to Questions

**Comments to the Author**

1. Is the manuscript technically sound, and do the data support the conclusions?

Reviewer #1: Partly

Reviewer #2: Partly

2. Has the statistical analysis been performed appropriately and rigorously? 

Reviewer #1: No

Reviewer #2: No

3. Have the authors made all data underlying the findings in their manuscript fully available?

Reviewer #1: Yes

Reviewer #2: No

4. Is the manuscript presented in an intelligible fashion and written in standard English?

Reviewer #1: Yes

Reviewer #2: Yes

5. Review Comments to the Author

Reviewer #1: In the submitted manuscript, the authors studied the infection rates, mortality, and hospitalizations from SARS-Coronavirus-2 disease in the state of Indiana from March 1, 2020, through December 31, 2020. In their analysis, the authors classified the disease outcomes based on race, age, gender, and residence designated by rural/urban status. The authors compared three phases based on social distancing and other state policies. The modality of data analysis is descriptive. I agree with the authors that longitudinal/time series analysis of Covid-19 is limited in the literature, however, emerging. I felt the analysis, the results, and the conjectures did not elevate the manuscript to a level with substantial innovation.

There is plenty of published work on the racial disparities and the effect of social determinants of health on Covid-19. Many of those works used rigorous data analytic approaches. I am including a few of them here:

1. Millett GA, Jones AT, Benkeser D, et al. Assessing differential impacts of COVID-19 on black communities. Ann Epidemiol. 2020;47:37-44.

2. DiMaggio C, Klein M, Berry C, Frangos S. Blacks/African Americans are 5 times more likely to develop COVID-19: spatial modeling of New York city ZIP code-level testing results. Ann Epidemiol. 2020;51:7-13

There is substantial work done assessing rural/urban differences as well, for example

1. Paul, R., Arif, A., Pokhrel, K., & Ghosh, S. (2021). The association of social determinants of health with COVID‐19 mortality in rural and urban counties. The Journal of Rural Health. https://doi.org/10.1111/jrh.12557

2. Karim, S. A., & Chen, H. F. (2021). Deaths From COVID‐19 in Rural, Micropolitan, and Metropolitan Areas: A County‐Level Comparison. The Journal of Rural Health, 37(1), 124-132.

3. Peters, D. J. (2020). Community susceptibility and resiliency to COVID‐19 across the rural‐urban continuum in the United States. The Journal of Rural Health, 36(3), 446-456.

Similar analyses presented in the submitted manuscript can be easily available on many state and county websites, for example, see Indiana (https://www.coronavirus.in.gov/2393.htm) and North Carolina (https://covid19.ncdhhs.gov/dashboard/cases-demographics).

Reviewer #2: This is important research that highlights rural and racial inequalities in COVID outcomes. However, the authors need to address several issues before this paper is publishable

The paper needs thorough editing to avoid clunky sentences. See example in line 57: There further exists little evidence on individuals in rural communities… There are also typos and other grammatical issues. Please proofread carefully.

What are the demographic characteristics of Indiana? This information can help you set the relevance of this paper. Particularly in reference to the % of the population living in rural areas and minority groups.

Methodologically, the analysis is largely descriptive. I advise the authors to run some regressions (or even t-tests and ANOVAs) to tell us if the differences between minorities v. whites and rural v. urban are statistically significant. Add those who classify themselves as others racially to the analysis and graphs since you have a hunch they are largely Hispanics. Add to the figures.

I don’t agree with the twindemic characterization. The race and geography risk should be framed as compounded risk factors rather than a twindemic. A twindemic implies two simultaneous health issues, like COVID and mental disorders.

In the limitations, when you say: The patterns observed in Indiana may not generalize to all geographic regions of the U.S. or other countries (276-277) - stress that these patterns might be more applicable to states with similar proportions of the population living in rurality and with comparable rates of minority populations (see second comment).

282 – Correct mitigation (it says mitgation) and say mitigation strategies because the sentence is repetitive: (282-283) “this suggests aggressive mitgation for a longer period of time may be necessary for stronger mitigation”.

286: I think there is a missing word here “As public health attempts to mitigate disease spread going forward”

302 – I think you mean you can achieve equity in vaccinating

306 – Higher morbidity and mortality (add higher)

6. PLOS authors have the option to publish the peer review history of their article (what does this mean?). If published, this will include your full peer review and any attached files.

Reviewer #1: No

Reviewer #2: No

---

## [Author Response · Author response to Decision Letter 0]

20 May 2021

We uploaded a separate document outlining our responses to the reviewers and the edits we made to the paper.

---

## [Decision Letter · Decision Letter 1]

9 Jul 2021

The synchronicity of COVID-19 disparities: Statewide epidemiologic trends in SARS-CoV-2 morbidity, hospitalization, and mortality among racial minorities and in rural America

PONE-D-21-05556R1

Dear Dr. Dixon,

We’re pleased to inform you that your manuscript has been judged scientifically suitable for publication and will be formally accepted for publication once it meets all outstanding technical requirements.

Kind regards,

Jeffrey Shaman

Academic Editor

PLOS ONE

Additional Editor Comments (optional):

Reviewers' comments:

Reviewer's Responses to Questions

**Comments to the Author**

1. If the authors have adequately addressed your comments raised in a previous round of review and you feel that this manuscript is now acceptable for publication, you may indicate that here to bypass the “Comments to the Author” section, enter your conflict of interest statement in the “Confidential to Editor” section, and submit your "Accept" recommendation.

Reviewer #1: All comments have been addressed

2. Is the manuscript technically sound, and do the data support the conclusions?

Reviewer #1: Partly

3. Has the statistical analysis been performed appropriately and rigorously? 

Reviewer #1: Yes

4. Have the authors made all data underlying the findings in their manuscript fully available?

Reviewer #1: No

5. Is the manuscript presented in an intelligible fashion and written in standard English?

Reviewer #1: Yes

6. Review Comments to the Author

Reviewer #1: I am pleased to see that the authors carefully considered the comments from the previous version. ANOVA tests helped in strengthening the manuscript. However, I still think the innovation and the additional information that we are getting from this article beyond the state website is very little.

7. PLOS authors have the option to publish the peer review history of their article (what does this mean?). If published, this will include your full peer review and any attached files.

Reviewer #1: No